# Bioactive Molecules from Mangrove *Streptomyces qinglanensis* 172205

**DOI:** 10.3390/md18050255

**Published:** 2020-05-13

**Authors:** Dongbo Xu, Erli Tian, Fandong Kong, Kui Hong

**Affiliations:** 1Key Laboratory of Combinatorial Biosynthesis and Drug Discovery, Ministry of Education, School of Pharmaceutical Sciences, Wuhan University, Wuhan 430071, China; xudongbo1990@gmail.com (D.X.); erlitian@whu.edu.cn (E.T.); 2Institute of Tropical Bioscience and Biotechnology, Chinese Academy of Tropical Agricultura Sciences, Haikou 571101, China; kongfandong@itbb.org.cn

**Keywords:** mangrove *Streptomyces*, genetic dereplication, anti-microbial, antiproliferative

## Abstract

Five new compounds 15*R*-17,18-dehydroxantholipin (**1**), (*3E*,*5E*,*7E*)-3-methyldeca-3,5,7-triene-2,9-dione (**2**) and qinlactone A–C (**3**–**5**) were identified from mangrove *Streptomyces qinglanensis* 172205 with “genetic dereplication,” which deleted the highly expressed secondary metabolite-enterocin biosynthetic gene cluster. The chemical structures were established by spectroscopic methods, and the absolute configurations were determined by electronic circular dichroism (ECD). Compound **1** exhibited strong anti-microbial and antiproliferative bioactivities, while compounds **2**–**4** showed weak antiproliferative activities.

## 1. Introduction

Microbial natural products are an important source of drug lead. Mangrove streptomycetes were reported as a potential source of plenty of antiproliferative or anti-microbial chemicals with novel structures [1]. The bioinformatics of easily available genome information from microorganisms breaks the bottleneck of traditional natural product discovery to a certain extent, and secondary metabolites isolation guided by genome sequence has increasingly become a research frontier [2]. Genome mining and silent gene cluster activation unveil the potential of diverse secondary metabolites in bacteria [3,4,5]. The OSMAC (One Strain Many Compounds) approach has been proven to be a simple and powerful tool to mine new natural products [6,7]. Due to complexity profiles of secondary metabolites including intermediates, a strategy named “genetic dereplication” was also developed to simplify the profiles by eliminating the major known secondary metabolites’ biosynthetic pathway, so that more easily detecting other novel compounds and/or reversing the precursor pools for other low expressed pathways in the microorganisms [8].

Previous studies reported that enterocin and its metabolites are the main and high-yield products in *Streptomyces qinglanensis* 172205 [9]. After the whole genome sequence was obtained, we analyzed the gene clusters of secondary metabolites, and found that more than 50% of them are coding for unknown compounds. However, enterocin was always detected in all of the media used during the OSMAC study. Hence, in this study, to mine the unknown compounds in strain 172205, we carried out the genetic dereplication strategy by which we deleted the enterocin biosynthetic gene cluster in genome and then detected the diversity of secondary metabolites profiles by OSMAC method. A mutant strain 172205Δ*enc* was generated by the whole enterocin biosynthetic gene cluster deletion using double-crossover homologous recombination and tested by HPLC fingerprint profiles for diverse products of crude extracts from 10 kinds of liquid fermentation media. The results showed that strain 172205Δ*enc* could produce the most diverse peaks on HPLC under fermentation in D.O. (dextrin-oatmeal) medium. Subsequently, a large-scale fermentation with D.O. medium was performed. After isolation and purification of the compounds from the crude extract, five new compounds including 15*R*-17,18-dehydroxantholipin (**1**), (3*E*,5*E*,7*E*)-3-methyldeca-3,5,7-triene-2,9-dione (**2**) and qinlactone A–C (**3**–**5**) were identified. Their structures were elucidated by one-dimensional (1D)/two-dimensional (2D) nuclear magnetic resonance spectroscopy (NMR) data, as well as electronic circular dichroism (ECD) calculation. In anti-microbial bioassay tests, compound **1** showed strong anti-*Staphylococcus aureus* and anti-*Candida albicans* activities with MIC (minimum inhibitory concentration) values of 0.78 µg/mL and 3.13 µg/mL, respectively. For antiproliferative bioactivity, compound **1** exhibited strong cytotoxicities against human breast cancer cell line MCF-7 and human cervical cancer cell line HeLa with IC_50_ values of 5.78 µM and 6.25 µM, respectively, while compound **2**–**4** showed weaker antiproliferative activities with IC_50_ values ranging from 129 to 207 µM. Therefore, the “genetic dereplication” strategy is useful to find compounds that synthesized by low expression gene clusters and would be of interest to colleagues in natural product discovery.

## 2. Results

Strain 172205Δ*enc* was obtained by the enterocin biosynthetic gene cluster deletion (Figure 1a) and confirmed by PCR (polymerase chain reaction) amplification (Appendix A). The HPLC profile of crude extract in D.O. medium (Figure 1b) showed that enterocin biosynthesis were totally blocked in mutant strain 172205Δ*enc*, including its intermediate metabolite-cinnamic acid.

Almost 80 grams crude extract was obtained from extraction of 60 L fermentation broth of mutant 172205Δ*enc* and subjected to column chromatography and semi-preparative HPLC purification to afford compounds **1**–**5.** The chemical structures were showed in Figure 2.

15*R*-17,18-dehydroxantholipin (**1**) was obtained as a dark red powder with the molecular formula of C_27_H_16_ClNO_9_ (HRESIMS-high resolution electrospray ionisation mass spectrometry m/z 534.0587, calcd 534.0586 for [M + H]^+^), implying 20 degrees of unsaturation. Detailed comparison of 1D NMR (Table 1) between compound **1** and the reported xantholipin [10,11] showed that **1** had the same NMR data with xantholipin, except for the two sp^2^ quaternary carbons C-17 (*δ*_C_ 136.0) and C-18 (*δ*_C_ 141.5), implying a double bond. The mass spectrum suggested the loss of a H_2_O group to form a double bond between C-17 and C-18. The correlations in HMBC (heteronuclear multiple bond coherence) from H-16a to C-13, C-15, C-17 and C-18 also located the placement at C-17 and C-18. Thus, the planar structure of **1** was determined. The key HMBC and COSY (homonuclear correlation spectroscopy) correlations are shown in Figure 3. Furthermore, the quantum chemical ECD calculation method was also used to determine the absolute configuration. The calculated ECD spectrum of **1** was compared with the experimental one, which revealed an excellent agreement between them (Figure 4). Therefore, the absolute configuration of **1** was assigned to 15*R*. Thus, the structure of **1** was determined.

(3*E*,5*E*,7*E*)-3-methyldeca-3,5,7-triene-2,9-dione (**2**) was obtained as a yellow powder. The molecular formula of **2** was determined as C_11_H_14_O_2_ (HRESIMS *m/z* 179.1064, calcd 179.1067 for [M + H]^+^), indicating 5 degrees of unsaturation. The 1D and HSQC (heteronuclear single quantum coherence) NMR data (Table 1) of **2** revealed the presence of three aliphatic methyls, two carbonyls (ketone) (*δ*_C_ 202.0 and 201.3), five olefinic methines (C-4 to C-8) and six olefinic carbons. The ^1^H-^1^H COSY correlations (Figure 3) from H-4 to H-8 revealed the conjugated system indicated by bold lines in Figure 3. HMBC correlations from H-5 to C-3 and C-7, H-6 to C-4 and C-7, H-7 to C-5 and C-6 and H-8 to C-6 and C-7 suggested that **2** contained three conjugated double bonds. HMBC correlations from H-11 to C-2 and C-3, and H-4 to C-2 and C-11 located one methyl (H-11) at C-3. HMBC correlations from H-1 to C-2 and C-4, and H-11 to C-2 and C-3 supported one methyl (C-1) located at C-2 and the connection between C-2 and C-3. Moreover, HMBC correlation from H-10 to C-8 and C-9, H-8 to C-9, and H-7 to C-9 assigned another methyl ketone location at C-8. *J*_H-5/H-6_ = 14.6 Hz and *J*_H-7/H-8_ = 15.6 Hz revealed two (*E*)-alkene between C-5-C-8. The ROESY (rotating frame overhauser effect spectroscopy) correlation of H-4 and H-1 supported the (*E*)-alkene between C-3 and C-4. Thus, the (3*E*,5*E*,7*E*)-triene was identified and chemical structure of **2** was established.

Qinlactone A (**3**) was obtained as a colorless oil. Its HRESIMS data indicated its molecular formula is C_16_H_22_O_4_ (*m/z* 279.1587, calcd 279.1591 for [M + H]^+^) with 6 degrees of unsaturation. The 1D (Table 2) and HSQC NMR data of **3** revealed the presence of five aliphatic methyl groups, one carbonyl (ketone, *δ*_C_ 202.2) and one carbonyl (ester, *δ*_C_ 183.3) and five olefinic methines (C-5 to C-9). The ^1^H-^1^H correlations of H-5/H-6/H-7 and H-8/H-9 confirmed the same three conjugated double bonds as **2**. The HMBC correlations (Figure 3) from H-12 to C-11 (*δ*_C_ 202.2) and C-10, H-9 to C-11 and H-16 to C-8/C-9/C-10/C-11 confirmed the location of one methyl ketone and a methyl (C-16) at C-10. ROESY correlations of H-12/H-9, H-8/H-16 and H-9/H-7 confirmed all the (E)-alkenes from C-5 to C-10. Additionally, HMBC correlations from H-13 to C-1/C-2/C-3/C-14 and H-14 to C-1/C-2/C-3/C-13 suggested two aliphatic methyl groups located at quaternary carbon C-2, and indicated the connection from C1 to C3. HMBC correlations from H-15 to C-3/C-4 located the last methyl (*δ*_C_ 22.1) at C-4. Two carbons connected to oxygen atoms (*δ*_C_ 81.5 and 88.0) suggested the OH group at C-3 and ester oxygen connected with C-4. Thus, based on the unsaturation and ester group (*δ*_C_ 183.8, C-1), a γ-lactone structure was revealed. The HMBC correlations from H-3 to C-1/C-2/C-4/C-5/C-13/C-14 also confirmed γ-lactone in **3**. Meanwhile, the HMBC correlations of H-5 to C-4 and H-6 to C-4 revealed the connection of lactone and conjugated olefin part. Thus, the planar structure of **3** was established. The relative configuration of **3** was established by ROESY experiment. The ROESY correlation of H-3 and H-14/H-5 suggested the relative configuration of *3R**, *4S** (Figure 3). The absolute configuration was confirmed by a good agreement between the calculated ECD spectrum of **3** and experimental one (Figure 4). Therefore, the absolute configuration **3** was assigned to *3R, 4S*. Compound **3** was named qinlactone A.

Qinlactone B (**4**) was obtained as a colorless oil and assigned the same molecular formula as **3** by HRESIMS (*m/z* 279.1587, [M + H]^+^). The 1D and 2D NMR data (Table 2) of **4** corresponded closely to those of **3**, which suggested **4** had the same planar structure with **3** as epimer instead of enantiomer. The ROESY correlations of H-3 and H-15/H-13 suggested the relative configuration of *3R**, *4R** (Figure 3). The absolute configuration was determined as *3R*, *4R* by the similar Cotton effects between the calculated ECD spectrum and experimental one of **4** (Figure 4). Thus, the structure of **4** was determined (Figure 2), named qinlactone B.

Qinlactone C (**5**) was obtained as a light-yellow oil with the molecular formula C_16_H_24_O_6_, determined by HRESIMS *m/z* 313.1650 (calcd 313.1646 for [M + H]^+^), implying 5 degrees of unsaturation. Compared with **3**, 1D NMR data (Table 2) of **5** revealed the absence of two olefinic methines, but the presence of two methines connected with oxygen atoms (*δ*_C/H_ 72.8/4.44 and 77.9/3.63). Based on the formula and unsaturation analysis, **5** contained a vicinal diol at C-5 and C-6, which was also confirmed by ^1^H-^1^H COSY correlations from H-5 to H-9 and HMBC correlations from H-5 to C-3 and C-4 (Figure 3). Therefore, the planar structure of **5** was identified. ROESY correlations of H-3 and H-13/H-5 suggested the relative configuration of *3R**, *4R** in the lactone ring (Figure 3). However, due to the existing vicinal diol structure, the relative and absolute configuration of **5** could not be determined based on the present data. Thus, compound **5** was named qinlactone C.

In the anti-microbial bioassay test, only compound **1** exhibited strong bioactivity against *Staphylococcus aureus* and *Candida albicans*, with MIC values of 0.78 µg/mL and 3.13 µg/mL, respectively. Meanwhile, in antiproliferative bio-test, **1** showed strong inhibitory effects on MCF-7 and HeLa cell lines with IC_50_ values of 5.78 µM and 6.25 µM, respectively (Table 3). Compound **2**–**4** showed weak activities against MCF-7 and HeLa cell lines with IC_50_ values ranging from 129 to 207 µM (Table 4).

## 3. Discussion

In this study, we identified five new compounds with bioactivities from mangrove *Streptomyces qinglanensis* 172205 with “genetic dereplication.” Compound **1** showed strong anti-*Staphylococcus aureus* and anti-*Candida albicans* activities with MIC values of 0.78 µg/mL and 3.13 µg/mL, respectively, and exhibited strong cytotoxicities against MCF-7 and HeLa cell lines with IC_50_ values of 5.78 µM and 6.25 µM, respectively. However, compound **2**–**4** exhibited only weak antiproliferative activities with IC_50_ values ranging from 129 to 207 µM.

Attempting to activate gene clusters of possible unknown secondary metabolites in strain 172205, we deleted the whole biosynthetic gene cluster of the main product enterocin. Through HPLC detection, we found that this strategy in strain 172205 did not clearly activate any new metabolites in several media. However, we still focused on some low-yield products, which produced in the wild type strain as well (Figure 1). However, the mutant strain without the main product enterocin facilitated detection and isolation of the low-yield products, from which we finally identified five new compounds. Thus, “genetic dereplication” did help simplifying the process of isolation and mining the low-yield products. This strategy would be more effective for identifying multiple types of metabolites in one strain, if combined with other genome mining tools or methods.

15*R*-17,18-dehydroxantholipin is an analog of reported xantholipin [10], which exhibited similar strong antiproliferative and anti-microbial activities. In fact, we firstly identified a gene cluster by antiSMASH analysis in the genome of strain 172205 (Appendix A), which had a high similarity with the reported xantholipin gene cluster [12]. Then, we tried at least 10 media to detect the similar UV absorption of xantholipin by HPLC and characterized mass for halogen compounds by HRESIMS, and finally detected analogs in products from D.O. medium, which is the same recipe with the reported medium to produce xantholipin. Compound with the targeted UV absorption was identified as 15*R*-17,18-dehydroxantholipin. Lacking the oxidoreductase gene *xanZ2* which was proposed for the double band reduction at C-17 and C-18 [12], resulted in the production of 15*R*-17,18-dehydroxantholipin in strain 172205. Hence, genome-guided compound discovery combined with OSMAC is an effective method for isolation and identification of some well-known and valuable compounds or potential new analogs. Moreover, multiple strategies of genome mining will be helpful to mine the potential of natural products in one strain.

## 4. Materials and Methods 

### 4.1. General Experimental Procedures

Ultraviolet (UV) spectra were recorded on a Shimadzu UV-2401 PC UV-Visible spectrophotometer. ECD spectra were recorded on an Applied PhotoPhysics Chirascan. IR spectra were recorded on Bruker Tensor27 spectrometer. Optical rotations were measured with JASCO P-1020. 1D and 2D NMR spectra were recorded in DMSO-*d*_6_ and CD_3_OD-*d*_4_ on Bruker DRX-500. Chemical shifts (*δ*) were expressed in ppm with reference to the solvent signals. High resolution mass spectra were recorded on a Thermo Instruments MS system (LTQ Orbitrap) equipped with a Thermo Instruments HPLC system with a Thermo Hypersil GOLD column (150 × 4.6 mm) with electrospray ionization in the positive-ion mode. Analytical HPLC was performed on a Waters 2998 with a photodiode array detector (PDA) detector with a Phenomenex Gemini (C18 250 × 4.6 mm) column. Semi-preparative HPLC was carried out on an Agilent 1260 Infinity with a diode array detector (DAD) with an Agilent Zorbax SBC18 (250 × 9.4 mm) column. Sephadex LH-20 (Shanghai kayon Biological Technology) was used for column chromatography.

### 4.2. Microorganism Material and Culture

*Streptomyces qinglanensis* 172205 was isolated from a mangrove soil sample and identified as a novel species [13]. This strain was cultured on ISP2 agar plates at 28 °C. *Escherichia coli* Top 10 was used for cloning and *E. coli* ET12567/pUZ8002 was used for intergeneric conjugation, which cultured on LB agar plates at 37 °C. 

### 4.3. Gene Cluster Deletion

Strain 177205 was reported to produce the main product enterocin and its biosynthesis gene cluster was located in genome [9]. Hence, strain 172205Δ*enc* was constructed by the whole enterocin biosynthetic gene cluster deletion with double-crossover homologous recombination. To construct the plasmid for gene cluster deletion, two DNA fragments, an 1870 bp *Avr* II-*Hin*d III homologous arm and another 1829 bp *Nde* I-*Hin*d III homologous arm were cloned from strain 17225 genome DNA covering both ends of the enterocin biosynthetic gene cluster. DNA fragments were ligated into pMD19-T simple and fragment sequences were identified by DNA sequencing. Two recycled DNA fragments were inserted into the delivery vector pYH7 [14] by *Nde* I-*Hin*d III restriction sites to yield pWHU2343 (Appendix A). Intergeneric conjugation of plasmid pWHU2343 into strain 172205 by *E. coli* ET12657/pUZ8002 were carried out as protocol described in Practical Streptomyces Genetics [15]. The donor ET12657/pUZ8002 containing plasmid and the recipient spores were mixed and spread on Mannitol-Soy-agar (MS) plates with 10 mM MgCl_2_ and grown for 14 h at 28 °C. Then the plates were overlaid with 1 mL sterile water contained 4 µg/mL apramycin and 25 µg/mL nalidixic acid. Single colonies were transferred to a new MS plate with same antibiotics for further confirmation of antibiotic resistance. To screen the double-crossover mutants, single clones from no antibiotics plate were replicated on a MS plates with apramycin. Genome DNA of all candidates that had no apramycin resistance were extracted for PCR identification. Four pair of primers, *enc*-U, *enc*-D, *enc*-UD and *enc*-M, were used for screening (Appendix A), and a specific 1080 bp product for *enc*-UD was only amplified in mutant clones with absence of 21.6 kb gene cluster, but specific products for *enc*-U, *enc*-D and *enc*-M were only amplified in genome of wild-type clones. The mutant was also confirmed by detecting enterocin production in fermentation.

### 4.4. Extraction and Isolation

Strain 172205Δ*enc* spores were inoculated into seed broth medium, cultured at 200 rpm, 28 °C for 3 days. Then seed broth was transferred to 200 of 1 L flasks consisted of 300 mL fermentation medium, shaken at 200 rpm for 7 days (media recipes in literature [12]). The broth was extracted by organic reagent as described [9] and evaporated to dryness (80 g). Samples and fractions were tested by HPLC fingerprint [9]. HPLC fingerprints were carried using the following gradient: H_2_O (A)/MeOH (B): 0 min, 10% B; 15 min, 100% B; 20 min, 100% B; 21 min, 10% B; 30 min, 10% B; flow rate of 1 mL/min.

The crude extract was subjected to column chromatography on silica gel eluted by PE:CH_2_Cl_2_ (gradient from 1:0, 1:1 and 0:1, v:v) and CH_2_Cl_2_:MeOH (gradient from 100:1, 50:1, 5:1, 2:1 to methanol, v:v) to give A–F fractions. Importantly, the mix of crude extract and silica gel on column after elution was extracted by DMSO. DMSO layers were mixed with equal volume of NaCl saturated solution, and extracted again with ethyl acetate, then evaporated to fraction G. Fraction G was dissolved in DMSO and purified by HPLC (MeOH: H_2_O = 75:25, flow rate 3 mL/min) to afford compound **1** (4 mg, *t_R_* = 23.2 min). Fraction D was subject to silica gel column chromatography using cyclohexane: acetone (20:1 to 0:1 v:v) to afford five subfractions (D1–D5). Fraction D3 was purified by HPLC (MeCN: H_2_O = 30:70) to afford compound **2** (15 mg, *t_R_* = 16.9 min). Fraction D2 was purified by HPLC (MeOH: H_2_O = 48:52) to afford compound **3** (5 mg, *t_R_* = 30.7 min) and **4** (4 mg, *t_R_* = 31.9 min). Fraction D5 was purified by HPLC (MeCN: H_2_O = 20:80) to afford compound **5** (6 mg).

15*R*-17,18-dehydroxantholipin (**1**): dark red powder; [α]${}_{D}^{20}$ −168.85 (c 0.046, MeOH/CHCl_3_ = 5:1); UV (MeOH) *λ*_max_ (log *ε*): 247 (4.34), 279 (4.23), 316 (4.08), 478 (3.80) nm; CD (c 0.041, CHCl_3_) *λ*_max_ (Δε): 231 (−7.47), 249 (+5.09), 278 (−5.78), 314 (−1.29), 352 (−6.10), 485 (+1.41) nm; IR (KBr) *υ*_max_ 3428, 2925, 1633, 1028 cm^−1^; ^1^H and ^13^C NMR data, see Table 1; positive ion HRESIMS *m/z* 534.0587 [M + H]^+^ (calcd for C_27_H_17_ClNO_9_, 534.0586).

(*3E*,*5E*,*7E*)-3-methyldeca-3,5,7-triene-2,9-dione (**2**): yellow powder; UV (MeOH) *λ*_max_ (log *ε*): 225 (3.80), 324 (4.96) nm; IR (KBr) *υ*_max_ 3430, 2955, 2930, 1709, 1664, 1361, 1253, 998, 604 cm^−1^; ^1^H and ^13^C NMR data, see Table 1; positive ion HRESIMS *m/z* 179.1064 [M + H]^+^ (calcd for C_11_H_15_O_2_, 179.1067).

Qinlactone A (**3**): colorless oil; [α]${}_{D}^{19}$ +31.55 (c 0.727, MeOH); UV (MeOH) *λ*_max_ (log *ε*): 223 (4.07), 311 (4.40) nm; CD (c 0.013, MeOH) *λ*_max_ (Δε): 200 (−0.6), 224 (+1.3), 316 (+0.4) nm; IR (KBr) *υ*_max_ 3437, 2983, 2936, 1756, 1640, 1386, 1273, 1224, 1056, 995 cm^−1^; ^1^H and ^13^C NMR data, see Table 2; positive ion HRESIMS *m/z* 279.1587 [M + H]^+^ (calcd for C_16_H_23_O_4_, 279.1591).

Qinlactone B (**4**): colorless oil; [α]${}_{D}^{19}$ −93.91 (c 0.553, MeOH); UV (MeOH) *λ*_max_ (log *ε*): 221 (4.11), 313 (4.46) nm; CD (c 0.011, MeOH) *λ*_max_ (Δε): 201 (+1.2), 225 (−2.9), 310 (−3.8) nm; IR (KBr) *υ*_max_ 3432, 2980, 2934, 1774, 1758, 1716, 1640, 1386, 1274, 1225,1113, 1065, 996, 954 cm^−1^; ^1^H and ^13^C NMR data, see Table 2; positive ion HRESIMS *m/z* 279.1587 [M + H]^+^ (calcd for C_16_H_23_O_4_, 279.1591).

Qinlactone C (**5**): light yellow oil; [α]${}_{D}^{20}$ −0.18 (c 0.552, MeOH); UV (MeOH) *λ*_max_ (log *ε*): 201 (3.81), 229 (3.78), 275 (4.04) nm; CD (c 0.055, MeOH) *λ*_max_ (Δε): 216 (−3.4), 270 (+0.4) nm; IR (KBr) *υ*_max_ 3415, 3194, 2986, 2942, 1758, 1678, 1401, 1285, 1205, 1137, 1066, 954, 837, 801, 723 cm^−1^; ^1^H and ^13^C NMR data, see Table 2; positive ion HRESIMS *m/z* 313.1650 [M + H]^+^ (calcd for C_16_H_25_O_6_, 313.1646).

### 4.5. Effect of Compounds on Anti-Microbial and Antiproliferative Bioactivities

Antiproliferative activities against HeLa and MCF-7 cell lines were evaluated by 3-(4,5-dimethylthiazol-2-yl)-2,5-diphenyl tetrazolium bromide (MTT) assay as described [9]. Briefly, 6000 cells were plated into 96 well plate cultured with 90 µL DMEM (Dulbecco’s Modified Eagle Medium) medium supplemented with 10% fetal bovine serum (FBS). After overnight culture, 10 µL compounds in 5% DMSO culture solutions with gradient final concentrations of 0.39, 0.78, 1.56, 3.13, 6.26, 12.5, 25 and 50 µg/mL were added into each well in triplicates, using positive control of paclitaxel. After 48 h incubation, 12 µL MTT solutions (final concentration of 0.5 mg/mL in PBS - phosphate buffered saline) were added and plates were further incubated for 4 h. Then, the medium was replaced gently by 100 µL DMSO. Plates were shaken and read by a Tecan Infinite M200 Pro reader at 570 nm, and the reference wavelength was 690 nm. The values of IC_50_ were calculated by GraphPad Prism 7.0 by applying nonlinear regression with (inhibitor) versus normalized response. 

Anti-microbial activities against *Escherichia coli* ATCC 25922, *Staphylococcus aureus* ATCC 51650 and *Candida albicans* ATCC 10231 were evaluated by microtiter broth dilution method as described [16,17] with some modifications. Briefly, 75 µL Lysogeny broth (LB) or yeast extract-peptone-dextrose (YPD) medium and 20 µL inoculums (5 × 10^5^ CFU/mL, colony-forming unit/mL) was plated into each well in 96 well plate. Then 5 µL test compounds with gradient final concentrations of 0.78, 1.56, 3.13, 6.26, 12.5, 25, 50 and 100 µg/mL was added into each well with three copies, using the positive control of kanamycin (for bacteria) and nystatin (for fungus). Plates were shaken at 200 rpm and cultured at 30 °C for 20 h. At last, plates were examined for bacteria growth by turbidity in daylight. The MICs were defined as the lowest concentration at which no microbial growth could be detected.

### 4.6. ECD Calculation

The calculations were performed using DFT on Gaussian 03, and the calculation details were follow strictly as described in literature [18]. Briefly, the calculations were performed by using the density functional theory (DFT) as carried out in the Gaussian 03 [19]. The preliminary conformational distribution search was performed using Frog2 online version [20]. Further geometrical optimization was performed at the B3LYP/6-31G(d) level. Solvent effects of methanol solution were evaluated at the same DFT level by using the SCRF/PCM method [21]. TDDFT [22,23,24] at B3LYP/6-31G(d) was employed to calculate the electronic excitation energies and rotational strengths in methanol, except compound **1** with calculation in chloroform.

## Figures and Tables

**Figure 1 marinedrugs-18-00255-f001:**
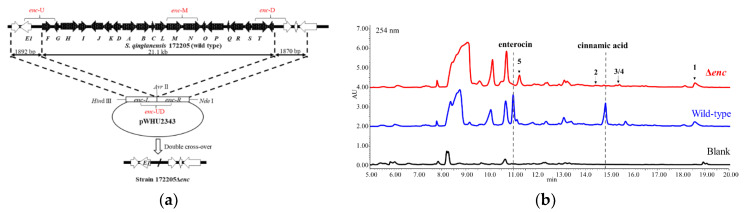
(**a**) The organization of enterocin biosynthetic gene cluster before and after deletion. Amplification fragments by verifying primers were labeled by fronts in red; (**b**) HPLC detection of crude extracts of strain 172205 wild type and Δ*enc* in D.O. medium.

**Figure 2 marinedrugs-18-00255-f002:**
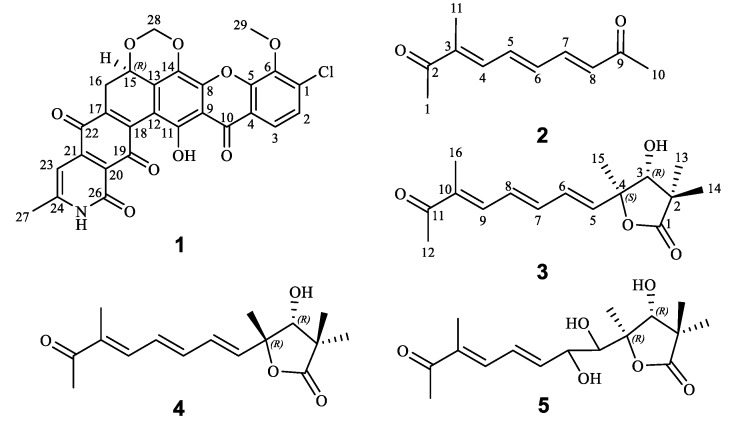
Chemical structures of compound **1**–**5**.

**Figure 3 marinedrugs-18-00255-f003:**
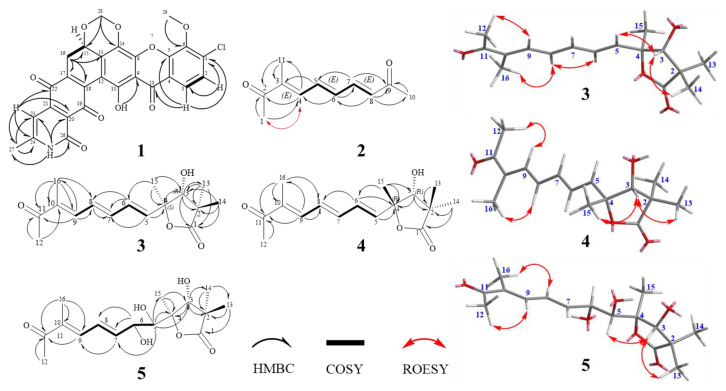
Key heteronuclear multiple bond coherence (HMBC), homonuclear correlation spectroscopy (COSY) and rotating frame overhauser effect spectroscopy (ROESY) correlations of compound **1**–**5**.

**Figure 4 marinedrugs-18-00255-f004:**
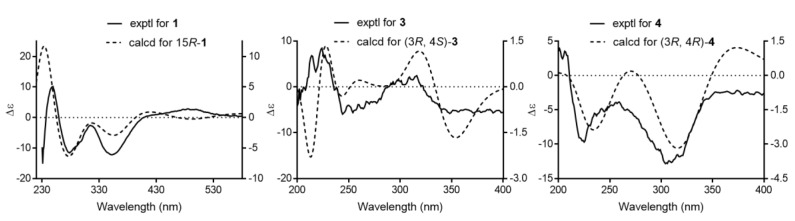
Experimental and calculated electronic circular dichroism (ECD) spectra for compound **1**, **3** and **4**.

**Table 1 marinedrugs-18-00255-t001:** ^1^H and ^13^C NMR data for Compound **1**–**2**.

Position	1			2		
	*δ*_H_ (*J* in Hz)	*δ*_C_, Type	HMBC	*δ*_H_ (*J* in Hz)	*δ*_C_, Type	HMBC
1		133.6, C		2.38 s	26.0, CH_3_	C-2, 4
2	7.63 d (8.8)	125.9, CH	C-1, 4, 6		202.0, C	
3	7.95 d (8.4)	120.8, CH	C-1, 5, 10		140.7, C	
4		120.6, C		7.28 d (11.4)	139.6, CH	C-2, 6, 11
5		149.7, C		7.16 dd (11.5, 14.6)	137.9, CH	C-3, 7
6		144.6, C		6.84 dd (11.2, 14.6)	138.8, CH	C-4, 7, 8
7				7.41 dd (11.1, 15.6)	144.5, CH	C-5, 6, 9
8		143.4, C		6.31 d (15.7)	133.7, CH	C-6, 9, 10
9		108.2 *, C			201.3, C	
10		181.4, C		2.32 s	27.4, CH_3_	C-7, 8, 9
11		158.9, C		1.94 d (1.0)	12.0, CH_3_	C-2, 3
12		109.5 *, C				
13		131.4, C				
14		131.3, C				
15	5.15 dd (6.4, 14.0)	71.1, CH				
16	a, 2.37 m	25.3, CH_2_	C-13, 15, 17, 18			
	b, 2.54 overlapped					
17		136.0 **, C				
18		141.5 **, C				
19		178.6, C				
20		117.3, C				
21		145.1, C				
22		182.8, C				
23	6.55 s	99.6, CH	C-20, 22, 24, 27			
24		154.5, C				
25-NH	12.67 s		C-20, 23, 26, 27			
26		151.9, C				
27	2.36 s	19.5, CH_3_	C-23, 24			
28	a, 5.71 d (5.9) b, 5.48 d (5.8)	91.3, CH_2_	C-14, 15 C-15			
29	4.08 s	61.6, CH_3_	C-6			
11-OH	12.67 s		C-9			

*, ** Assignments are made in comparison with literature data for similar reported compounds. **1** (800 and 200 MHz, DMSO-*d_6_*, δ in ppm); **2** (500 and 125 MHz, CD_3_OD-*d_4_*, δ in ppm).

**Table 2 marinedrugs-18-00255-t002:** ^1^H and ^13^C NMR data for Compound **3**–**5** (500 and 125 MHz, CD_3_OD-*d_4_*, *δ* in ppm).

Position	3			4			5		
	*δ*_H_ (*J* in Hz)	*δ*_C_, Type	HMBC	*δ*_H_ (*J* in Hz)	*δ*_C_, Type	HMBC	*δ*_H_ (*J* in Hz)	*δ*_C_, Type	HMBC
1		183.3, C			182.6, C			183.4, C	
2		45.9, C			44.9, C			45.0, C	
3	3.96 s	81.5, CH	C-1, 2, 4, 5, 13, 14	3.98 s	83.8, CH	C- 2, 4, 5, 13, 14	4.53 s	75.1, CH	C-2, 4, 5, 14, 15
4		88.0, C			86.2, C			90.4, C	
5	6.14 d (15.4)	142.1, CH	C- 3, 4, 7, 15	6.27 d (15.6)	139.0, CH	C- 4, 6, 7, 15	3.63 d (2.3)	77.9, CH	C-3, 4, 7, 15
6	6.46 dd (9.9, 15.4)	129.8, CH	C-4, 7, 8	6.46 dd (9.8,15.6)	130.2, CH	C-4, 5, 8	4.44 d (5.8)	72.8, CH	C-7, 8
7	6.71 overlapped	140.7, CH	C-6	6.71 overlapped	141.0, CH	C-8, 9	6.35 dd (5.6, 15.3)	144.4, CH	C-6, 9
8	6.71 overlapped	130.6, CH	C-6, 7	6.71 overlapped	130.4, CH		6.78 dd (11.2, 15.1)	127.9, CH	C-6, 9, 10
9	7.23 dd (1.0, 10.3)	141.3, CH	C-6, 8, 11, 16	7.23 dd (1.0, 10.0)	141.5, CH	C-7, 8, 11, 16	7.22 d (11.1)	141.1, CH	C-7, 8, 11, 16
10		137.7, C			137.5, C			137.3, C	
11		202.2, C			202.3, C			202.5, C	
12	2.34 s	25.8, CH_3_	C-9, 10, 11	2.33 s	25.8, CH_3_	C-9, 10, 11	2.34 s	25.8, CH_3_	C-9, 10, 11
13	1.19 s	20.5, CH_3_	C-1, 2, 3, 14	1.21 s	25.3, CH_3_	C-1, 2, 3, 14	1.24 s	25.6, CH_3_	C-1, 2, 3, 14
14	1.22 s	26.3, CH_3_	C-1, 2, 3, 13	1.04 s	19.9, CH_3_	C-1, 2, 3, 13	1.20 s	21.2, CH_3_	C-1, 2, 13
15	1.46 s	22.1, CH_3_	C-3, 4, 5, 6	1.52 s	27.4, CH_3_	C-3, 4, 5, 6	1.43 s	19.3, CH_3_	C-3, 4, 5,
16	1.87 d (1.0)	11.7, CH_3_	C-8, 9, 10, 11	1.87 d (0.8)	11.7, CH_3_	C-8, 9, 10, 11	1.87 s	11.6, CH_3_	C-9, 10, 11

**Table 3 marinedrugs-18-00255-t003:** MIC (μg/mL) against pathogenic microbes of compounds **1**–**5**. *E. coli*: *Escherichia coli*; *S. aureus*: *Staphylococcus aureus*; *C. albicans*: *Candida albicans*.

Compound.	*E. coli*	*S. aureus*	*C. albicans*
**1**	>100	0.78	3.13
kanamycin	6.25	6.25	/
nystatin	/	/	3.13

**Table 4 marinedrugs-18-00255-t004:** Cytotoxicities against MCF-7 and HeLa cells of compound **1**–**5** (µM).

Compound	MCF-7 (IC_50_ ± SD, 48 h)	HeLa (IC_50_ ± SD, 48 h)
**1**	5.78 ± 0.26	6.25 ± 0.29
**2**	206.91 ± 9.69	183.03 ± 11.11
**3**	>179.86	168.13 ± 13.15
**4**	136.87 ± 10.67	129.14 ± 3.98
Paclitaxel	<0.46	<0.46

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
