# Peer review of "Bioactive Molecules from Mangrove Streptomyces qinglanensis 172205"

_marinedrugs, 2020, doi:10.3390/md18050255_

Round 1
Reviewer 1 Report
This manuscript deals with the identification of five new compounds from a mangrove Streptomyces strain in which the enterocin biosynthetic gene cluster was deleted. This manuscript is recommended to be published in Marine Drugs after following points are revised.
- This manuscript should be edited by a native English speaker.
- Page 2, Figure 1: The observation of a peak of 1 in the HPLC chart for wild type should be discussed in the main text.
- Page 3, line 89: The HMBC correlation from H-1 to C-3 should be shown to assign the connectivity of C-2 to C-3.
- Pages 3 and 4: This reviewer feels that the ROESY assignments of 3–5 are rough. The relative configurations of 3–5 could not been assigned by ROESY correlations depicted in Figure 3. The 3D models of the lactone moieties of 3–5 should be shown, and careful assignment of the ROESY correlations of 3–5 should be carried out.
- Page 4, Figure 4: The calculated spectra of 3 and 4 are not similar to the experimental spectra.
- Throughout the manuscript: “anti-cancer” should be “antiproliferative”
Author Response
Response to Reviewer 1 Comments
Point 1: This manuscript should be edited by a native English speaker.
Response 1: Sorry for our language problem of our manuscript. We have proofread and revised again, do our best.
Point 2: Page 2, Figure 1: The observation of a peak of 1 in the HPLC chart for wild type should be discussed in the main text.
Response 2: Thanks. We have added a discussion part (Line 273) and taken this in.
Point 3: Page 3, line 89: The HMBC correlation from H-1 to C-3 should be shown to assign the connectivity of C-2 to C-3.
Response 3: In fact, we didn’t get the HMBC from H-1 to C-3. However, we get H-11 to C-2 and C-3. So we changed the statement into “HMBC correlations from H-1 to C-2 and C-4, and H-11 to C-2 and C-3 supported one methyl (C-1) located at C-2 and the connection between C-2 and C-3.”(Lines 90)
Point 4: Pages 3 and 4: This reviewer feels that the ROESY assignments of 3–5 are rough. The relative configurations of 3–5 could not been assigned by ROESY correlations depicted in Figure 3. The 3D models of the lactone moieties of 3–5 should be shown, and careful assignment of the ROESY correlations of 3–5 should be carried out.
Response 4: Thanks for your suggestion. We have added the 3D models of 3-5 with ROESY with key correlations. (Line 142)
Point 5: Page 4, Figure 4: The calculated spectra of 3 and 4 are not similar to the experimental spectra.
Response 5: All right. Now we have changed all right and left Y axis and put them in same level (make “0” in same level), right now much similar. (Line 145)
Point 6: Throughout the manuscript: “anti-cancer” should be “antiproliferative”
Response 6: Have changed in whole manuscript. Thanks.
Reviewer 2 Report
In the current communication, Dr. Hong and coworkers have reported the isolation, structure elucidation and bioactivity of five new molecules from mangrove derived Streptomyces qinglanensis 172205 by following the strategy of “genetic dereplication” to delete the enterocin biosynthetic gene cluster in the genome to simplify the secondary metabolite profile and facilitate the purification of less-abundant constituents of the extract. The author in his previous publications reported the isolation of enterocin with novel bioactivities from the same strain, Streptomyces qinglanensis 172205. The current manuscript reports the investigations on mutant strain 172205Δenc. The experiments and supporting data are adequate to support the claims. The manuscript could be accepted for publication after the authors have rectified the following major issues.
- Remove the word “New” from the title. Presumably this is why the manuscript is being published at all. The title of the manuscript should, perhaps, be changed to “Bioactive Molecules from a Mangrove-Derived Streptomyces strain”
- The introduction could be rendered much more effective by including the recent literature on anticancer compounds derived from Streptomyces like ACS Chem. Biol. 2020, 15, 3, 780-788 and Microorganisms 2019, 7(10), 394.
- In page 2, line 52, the authors have used terminology “anti-MCF7”and “anti-Hela” which are not appropriate and should be changed to cytotoxicity against MCF7…. Further the “Hela” should be written as “HeLa” at multiple places across the manuscript.
- In page 2, Figure 1b, the HPLC report shows the retention times for the enterocin and compound 5 are very close to each other. The authors should magnify the image which would allow better viewing for the readers. Further the authors need to specify briefly the conditions employed for the HPLC separation in the figure caption.
- In page 3, line 73, the term SP2 should be replaced with lower cases sp2
- The authors need to correct the C13 values throughout the manuscript. The chemical shift values should be rounded off to single digit after the decimal. For instance, page 3, line 73, δC for C-17 and C-18 should be rounded off to δC 136.0 and 141.5. In line 84, The δC for two ketones should be rounded off to δC 202.0 and 201.3
- In page 3, line 88, Compound 2, the authors need to recheck the HMBC correlation of H-4 to C-2 and C-4 and should be corrected as H-4 to C-2 and C-11.
- In multiple cases such as in line 89, 100 the δC values should be rounded off.
- In line 103, the HMBC correlations from H-13 to C-1/C-2/C-3-C-14 should be changed to H-13 to C-1/C-2/C-3/C-14.
- In page 4, line 127, for compound 5, the authors wrongly assigned the position of vicinal diol as C-4 and C-5 and needs to be changed as C-5 and C-6.
- In page 4, figure 3, compound 1, the methoxy carbon on aromatic ring should be numbered as 29.
- In Table 1, Line 4, compound 2, the authors represented a doublet at δ7.28 with coupling constant values (J = 0.9, 11.4) which is not possible. The author should recheck the proton NMR and report the correct coupling constant values.
- In Table 1, the authors should recheck the 1H and 13C values for compound 2. For instance, according to figure 2/3, the C-9 is ketone and C-10 is methyl group attached to ketone, but according to Table 1, H-9 resonates at δ6.31 (d, 1H, 15.7) and C-9 at δ133.70. There is no chemical shift value for H-10 and C-10 resonates at δ201.25, which cannot be true. The author missed the atom 7 in numbering in the Table 1 (because it’s O in compound 1, but not in compound 2), which confused the numbers of subsequent atoms in compound 2. The author should recheck both 1H and 13C spectra to report the correct values.
- In Table 1&2, the 13C chemical shifts for compound 1, 2, 3, 4, 5 should be rounded off to one digit after decimal.
- In Table 1&2, for compound 1, 2, 3, 4, 5, the authors should report the the state of carbons as CH, CH2, CH3 and C (from DEPT spectra). This will make Table 1 better organized for the readers.
- In page 5, line 146, the “CD3-OD-d4”should be replaced with “CD3-OD-d4”.
- In Table 2, compound 3, the author reported a “dd” at δ7.23 but gave only J1. The author should check the 1HNMR and report the correct coupling constant values.
- In Table 3 & 4, the authors reported the bioactivity of the isolated compounds for their antimicrobial and anticancer activity, but the standard error is not provided with the MIC and IC50 values. The authors should perform three biological replicates and provide the MIC and IC50 with SE if possible, or indicate that only a single measurement was performed.
- The IC50 values should be converted to molar concentrations (uM) at least for cancer cell line toxicity, and preferably for MIC, even if the field still accepts the mg/ml.
- The compounds 1 & 2 are solids, so melting points should be recorded.
- In SI, for all compounds, the 2D NMR spectra like HSQC, HMBC and COSY should be modified by assigning the proton and carbon number to the peaks and should represent the major correlations. Further the structure of the compounds should be included for the ease of viewing by the readers.
- In the SI, for all compounds, the author should represent the molecular formula, calculated molecular weight along with mass error in HRESIMS
Author Response
Response to Reviewer 2 Comments
Point 1: Remove the word “New” from the title. Presumably this is why the manuscript is being published at all. The title of the manuscript should, perhaps, be changed to “Bioactive Molecules from a Mangrove-Derived Streptomyces strain”
Response 1: Thank you for your good suggestion. We have removed “New”. And strain 172205 is an identified streptomyces, so may keep the strain name and people can get more info from the title or for easily search.
Point 2: The introduction could be rendered much more effective by including the recent literature on anticancer compounds derived from Streptomyces like ACS Chem. Biol. 2020, 15, 3, 780-788 and Microorganisms 2019, 7(10), 394.
Response 2: Already cited all in revised version.
Point 3: In page 2, line 52, the authors have used terminology “anti-MCF7”and “anti-Hela” which are not appropriate and should be changed to cytotoxicity against MCF7…. Further the “Hela” should be written as “HeLa” at multiple places across the manuscript.
Response 3: Already corrected them in whole manuscript.
Point 4: In page 2, Figure 1b, the HPLC report shows the retention times for the enterocin and compound 5 are very close to each other. The authors should magnify the image which would allow better viewing for the readers. Further the authors need to specify briefly the conditions employed for the HPLC separation in the figure caption.
Response 4: Already magnify the HPLC figure (Line 62) and give the detailed condition in method part 2.4 (Line 205).
Point 5: In page 3, line 73, the term SP2 should be replaced with lower cases sp2
Response 5: Corrected accordingly.
Point 6: The authors need to correct the C13 values throughout the manuscript. The chemical shift values should be rounded off to single digit after the decimal. For instance, page 3, line 73, δC for C-17 and C-18 should be rounded off to δC 136.0 and 141.5. In line 84, The δC for two ketones should be rounded off to δC 202.0 and 201.3
Response 6: Already changed in whole manuscript.
Point 7: In page 3, line 88, Compound 2, the authors need to recheck the HMBC correlation of H-4 to C-2 and C-4 and should be corrected as H-4 to C-2 and C-11.
Response 7: Right. Have corrected accordingly.
Point 8: In multiple cases such as in line 89, 100 the δC values should be rounded off.
Response 8: Already changed in whole manuscript.
Point 9: In line 103, the HMBC correlations from H-13 to C-1/C-2/C-3-C-14 should be changed to H-13 to C-1/C-2/C-3/C-14.
Response 9: Already revised.
Point 10: In page 4, line 127, for compound 5, the authors wrongly assigned the position of vicinal diol as C-4 and C-5 and needs to be changed as C-5 and C-6
Response 10: Already changed.
Point 11: In page 4, figure 3, compound 1, the methoxy carbon on aromatic ring should be numbered as 29.
Response 11: Already changed.
Point 12: In Table 1, Line 4, compound 2, the authors represented a doublet at δ7.28 with coupling constant values (J = 0.9, 11.4) which is not possible. The author should recheck the proton NMR and report the correct coupling constant values.
Response 12: Sorry for the confusion, we checked the H NMR and remove the 0.9 already.
Point 13: In Table 1, the authors should recheck the 1H and 13C values for compound 2. For instance, according to figure 2/3, the C-9 is ketone and C-10 is methyl group attached to ketone, but according to Table 1, H-9 resonates at δ6.31 (d, 1H, 15.7) and C-9 at δ133.70. There is no chemical shift value for H-10 and C-10 resonates at δ201.25, which cannot be true. The author missed the atom 7 in numbering in the Table 1 (because it’s O in compound 1, but not in compound 2), which confused the numbers of subsequent atoms in compound 2. The author should recheck both 1H and 13C spectra to report the correct values.
Response 13: so sorry for this confusion, and we have tried to make compound 1 and 2 in one figure, thing is I did miss the position 7 in Table 1, just copied in, resulting the wrong order for compound 2. And now we have corrected. Thanks for found this big mistake.
Point 14: In Table 1&2, the 13C chemical shifts for compound 1, 2, 3, 4, 5 should be rounded off to one digit after decimal.
Response 14: Already changed.
Point 15: In Table 1&2, for compound 1, 2, 3, 4, 5, the authors should report the the state of carbons as CH, CH2, CH3 and C (from DEPT spectra). This will make Table 1 better organized for the readers.
Response 15: Sound reasonable, and we already changed.
Point 16: In page 5, line 146, the “CD3-OD-d4”should be replaced with “CD3-OD-d4”.
Response 16: Already changed.
Point 17: In Table 2, compound 3, the author reported a “dd” at δ7.23 but gave only J1. The author should check the 1HNMR and report the correct coupling constant values.
Response 17: there was “1, 10.3” there. Already changed into “1.0, 10.3”, easy for reading.
Point 18: In Table 3 & 4, the authors reported the bioactivity of the isolated compounds for their antimicrobial and anticancer activity, but the standard error is not provided with the MIC and IC50 values. The authors should perform three biological replicates and provide the MIC and IC50 with SE if possible, or indicate that only a single measurement was performed.
Response 18: Actually, did it with three copies for MIC and IC50. MIC is just determined by observation without growth, and copies have same MIC value, not by software calculation. IC50 were calculated by GraphPad, and already added with SD. Thanks.
Point 19: The IC50 values should be converted to molar concentrations (uM) at least for cancer cell line toxicity, and preferably for MIC, even if the field still accepts the mg/ml.
Response 19: have convert IC 50 ug/ml into uM.
Point 20: The compounds 1 & 2 are solids, so melting points should be recorded.
Response 20: That’s right. We forgot this measure, however, we couldn’t get this data probably for now, because we basically ran out of our compound after several measurement and bio-tests.
Point 21: In SI, for all compounds, the 2D NMR spectra like HSQC, HMBC and COSY should be modified by assigning the proton and carbon number to the peaks and should represent the major correlations. Further the structure of the compounds should be included for the ease of viewing by the readers.
Response 21: Good suggestion. Have attached the chemical structure for each spectrum. We know that it’s much easier for readers if assigning some H and C number in SI spectrum. But this time, if possible, we would not do that and take more time for other revisions. And we think we have good description and summary of 1D NMR data, and especially for HMBC in main-text table. We hope you may agree with us. Thanks.
Point 22: In the SI, for all compounds, the author should represent the molecular formula, calculated molecular weight along with mass error in HRESIMS
Response 22: Good suggestion. Already added.
Reviewer 3 Report
General remarks
This manuscript contains a study about five new compounds were identified from mangrove Streptomyces qinglanensis 172205 with “genetic dereplication”.
The work is interesting, the results are well presented, but there is no discussion in this manuscript. For this reason, I can't in detail evaluate this manuscript.
Specific comments
1. How did the authors obtain the value of MIC 3.13ug/ml.
The MIC values resulting from the use of the reference method are as follows: 256, 128, 64, 32, 16, 8, 4, 2, 1, 0.5 ug/ml e.t.c.
If the determined value (3.13) is an average value, please calculate the standard deviation.
- The biological study is described in laconic. In my opinion, they require improvement in terms of methodology and discussion of the obtained results.
- The conclusions should include information on the biological activity of the presented compounds.
Author Response
Response to Reviewer 3 Comments
Point 1: The work is interesting, the results are well presented, but there is no discussion in this manuscript. For this reason, I can't in detail evaluate this manuscript
Response 1: Thanks for your great suggestion. We have added some discussion part in the end (Line 273).
Point 2: How did the authors obtain the value of MIC 3.13ug/ml. The MIC values resulting from the use of the reference method are as follows: 256, 128, 64, 32, 16, 8, 4, 2, 1, 0.5 ug/ml e.t.c. If the determined value (3.13) is an average value, please calculate the standard deviation.
Response 2: Sorry for the confusion. Actually, we used the different stock concentration with 2, 1, 0.5, 0.25…. mg/ml, compared to the ref. Then added into 96 well plate with 20X dilution. And MIC were determined by the lowest concentrations which totally inhibited the growth of the testing straining. We have put more detailed in method 2.5 (Line 240), thanks.
Point 3: The biological study is described in laconic. In my opinion, they require improvement in terms of methodology and discussion of the obtained results.
Response 3: We have put more details in the methods part (Line 240) and added some discussion in the end (Line 273).
Point 4: The conclusions should include information on the biological activity of the presented compounds.
Response 4: OK. Have added in. (Line 269)
Round 2
Reviewer 3 Report
1. I think it would be better to put the discussion together with the results. Results and discussion, not conclusion and discussion. Alternatively, discussion and conclusion.
2. Some abbreviations used at work have not been explained, eg. COSY, HMBC, MIC, MCF-7, MTT, D.O. DMEM, FBS, YPD.
Author Response
Thanks for your constructive comments. Please see the attachment for the response.
